# ImpResDescan: Diffusion-Based Restoration for Scanned Document Images via Implicit and Ambient Training

## Abstract

We propose **ImpResDescan**, a distortion-aware descanning framework that restores high-quality digital images from scans degraded by nonlinear color shifts, local artifacts, and geometric misalignments introduced by print–scan pipelines. Unlike the latest work, i.e., DescanDiffusion+, which applies linear channel-wise distribution correction and assumes perfect alignment, ImpResDescan removes those handcrafted assumptions through two components: (i) an *implicit color correction module* that couples a global encoder with a pixel-wise implicit mapper to learn a scan-conditioned, per-pixel nonlinear color transformation directly from data; and (ii) a *residual local refinement module* trained with an *ambient strategy* that is robust to spatial and semantic misalignment by supervising only keypoint-aligned regions and regularizing global structure with Multiscale Sliced Wasserstein Distance (MS–SWD). The *residual local refinement module* is additionally conditioned on a degradation-aware encoder, enabling robust removal of localized artifacts with low computational overhead and delivering **up to $2\times$ faster** inference than DescanDiffusion+. Extensive experiments across multiple datasets including **DESCAN-18K** (18,000 scan-original pairs) show that **ImpResDescan** consistently outperforms related restoration models in both fidelity and perceptual quality. On **DESCAN-18K**, it surpasses DescanDiffusion+ by $+0.9\,\mathrm{dB}$ PSNR, $-3.43$ FID, and $-18.6\%$ LPIPS.

## 1 Introduction

*Descanning* reconstructs scans to match their original digital images Cha et al. (2024), reversing degradations such as nonlinear color drift, noise, halftoning, texture distortions, and bleed-through. With large-scale digitization e.g., over 25M books in Google's Project Ocean Love (2017) and archival collections like the Digital Comic Museum D. C. Museum (2022) much content exists only as imperfect scans, underscoring the need for robust restoration.

Conventional image-restoration models typically assume globally uniform noise with single or few types of degradation Wang et al. (2018); Zamir et al. (2022); Zhu et al. (2017), thus struggling with highly varied and spatially complex distortions observed in real-world scans. By contrast, document-restoration systems exemplified by DocRes Zhang et al. (2024) target dewarping, deshadowing, and binarization rather than descanning, are trained largely on synthetic data, and do not reconstruct the pre-print digital image from print–scan inputs. DescanDiffusion+ Cha et al. (2024), a recent descanning method, addresses these issues partially through a two-stage pipeline: global affine color correction followed by local diffusion-based refinement. However, two key limitations remain. First, per-channel scale-and-shift corrections alone are insufficient, as printers and scanners introduce nonlinear, intensity-dependent color drifts, causing distinct error biases at different intensity levels. Second, DescanDiffusion+ assumes perfect spatial alignment between scans and their digital counterparts, resulting in sensitivity to modest spatial and semantic misalignments and thus leaving residual artifacts and softened details.

To overcome these limitations, we propose **ImpResDescan**, a novel distortion-aware descanning framework designed for robust restoration of images degraded by complex and varied scanning artifacts. Unlike previous approaches, our proposed method eliminates handcrafted priors through two

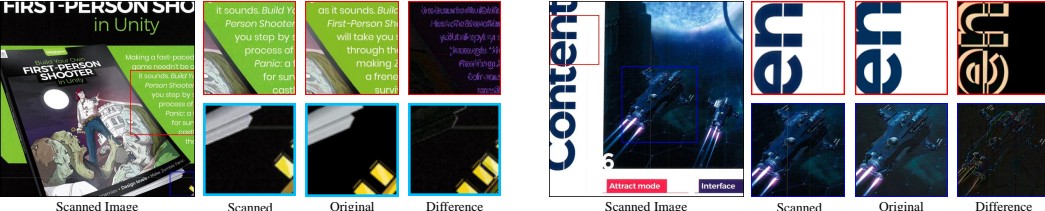

Figure 1: Each panel shows a scanned image, the corresponding crops from the scan and its digital original, and the per-pixel difference. Differences are dominated by **misalignment**: (1) **content misalignment** *actual content differences* between the scan and the original (e.g., different text); and (2) **spatial misalignment** small pixel-level shifts. Because misalignment residuals are typically much larger than actual degradation residuals, the raw difference mostly reflects misalignment and can hinder learning.

key innovations: 1) an *Implicit Color Correction Module*, where a global encoder combined with a pixel-wise implicit mapping function learns the nonlinear color transfer function directly from data, implicitly capturing intensity-dependent color biases without manual intervention; and 2) a *Degradation-Aware Refinement Stage*, explicitly correcting local degradations, including external noise, internal scanner noise, halftone patterns, texture distortions, and bleed-through effects, even under spatial and semantic misalignment. A dedicated distortion aware encoder conditions the refinement by providing detailed information about the overall distortion present in the image, while a residual diffusion-based decoder iteratively refines image restoration through incremental denoising steps.

During training, robustness to misalignment is enhanced by an ambient training strategy guided by keypoint matching Shen et al. (2024b). Local correspondences between matched keypoints are converted into a supervision mask, guiding the refinement process effectively even under substantial spatial misalignments. Additionally, the refinement stage is regularized using the Multiscale Sliced Wasserstein Distance (MS-SWD) He et al. (2024) loss, which promotes global visual consistency while preserving accurate local details.

Comprehensive evaluations on the **DESCAN-18K** dataset Cha et al. (2024), which contains 18,000 real scan–original pairs, show that **ImpResDescan** consistently outperforms prior methods including DescanDiffusion+ on both reference and non-reference quality metrics, establishing a new state of the art for practical descanning. Additional experiments on **Digital Comic Museum (Comic)** D. C. Museum (2022) and **DPS** Ho & Zhou (2022) further confirm these results: across multiple non-reference metrics, **ImpResDescan** surpasses baselines.

Our main contributions are:

- We propose **ImpResDescan**, a novel distortion-aware descanning framework that effectively addresses complex nonlinear color shifts, diverse local artifacts, and geometric misalignments found in scanned images.

- We introduce an **Implicit Color Correction Module** capable of learning nonlinear color transformations directly from data, eliminating the need for handcrafted calibration or simplistic affine corrections.

- We develop a robust **Degradation-Aware Refinement Stage**, employing a dedicated distortion-aware encoder and a residual diffusion-based decoder to correct local degradations including noise, halftone patterns, texture distortions, and bleed-through effects even under substantial spatial and semantic misalignments, guided by keypoint-based ambient supervision and regularized by MS-SWD.

- Through comprehensive experiments on **DESCAN-18K**, **Digital Comic Museum (Comic)**, and **DPS**, we show that **ImpResDescan** consistently surpasses state-of-the-art methods including DescanDiffusion+ in detail reconstruction, robustness, and overall image quality.

## 2 RELATED WORKS

### 2.1 DESCANNING

Descanning is the task of restoring images that suffer from a complex mixture of various degradations. These include color-related degradations (CD) and non-color-related degradations (NCD). This composite nature makes the problem challenging to solve with existing methods designed for restoring single or a few types of degradation Wang et al. (2018); Zhu et al. (2017); Zamir et al. (2022); Wan et al. (2020); Ho & Zhou (2022); Yu et al. (2022); Chen et al. (2021). Unlike descanning, document-restoration frameworks (e.g., DocRes Zhang et al. (2024)) address dewarping, deshadowing, and binarization and, being trained largely on synthetic data, are incapable of recovering the pre-print digital image from print-scan inputs.

DescanDiffusion+ Cha et al. (2024) is the first study to tackle the descanning problem. It specifically defined six types of distortions, classifying them into CD and NCD, and proposed a two-stage sequential algorithm consisting of a color correction module and a local texture refinement module. The color correction module restores the CD by predicting the mean and variance of the original R, G, B channels and applying a linear transformation (scale and bias) to the distorted image. Subsequently, the local texture refinement module takes the color-corrected image as a condition and feeds it into a Denoising Diffusion Probabilistic Model (DDPM) Ho et al. (2020) to restore the NCD. However, a major limitation of conventional diffusion models is the prohibitively long sampling process, as they start from pure Gaussian white noise. To address this, recent methods in the field of image restoration have shown promising results by utilizing residual images Yue et al. (2025); Shi et al. (2024). Therefore, we also adopt a residual diffusion model in our work.

### 2.2 NONLINEAR RELATION BETWEEN ILLUMINATION AND PIXEL INTENSITY

Modern digital imaging systems consist of a lens, a sensor, and an image signal processing (ISP) pipeline. This process inherently introduces a complex nonlinearity between the physical light intensity and the final digital pixel values Bae & Kim (2015). In a scanner system, light from a source is reflected linearly in proportion to the local reflectance of the document surface. This light is then linearly accumulated as charge at the sensor during the exposure time.

However, a significant nonlinearity is introduced by the scanner response function (similar to a camera response function Morovič (2008)) when this analog illumination signal is converted into the pixel intensity domain. DescanDiffusion+ Cha et al. (2024), however, overlooks this aspect by assuming a linear relationship. It performs color correction using only a single scaling and shift parameter, which results in suboptimal performance, especially in relatively bright or dark regions. To address this limitation, this paper proposes an implicit color correction module that is designed to account for this nonlinearity. Color distortion-intensity analysis are included in supplementary material.

### 2.3 MISALIGNMENT IN REALWORLD DATASETS

Image-to-image translation works best when the input and its clean target are well aligned. In real data, perfect alignment is rare. In descanning, we frequently observe both global and local misalignments, including pixel-level shifts and content-level changes (Fig. 1). DescanDiffusion+ Cha et al. (2024) tried to mitigate this by filtering misaligned pairs during the curation of DESCAN-18K. This reduces large global shifts, but many subtle local misalignments remain. These spatial inconsistencies add extra loss signals that distract the model from removing the actual distortions.

In this paper, we are the first to formally identify and tackle this critical issue in the descanning task. We introduce two novel components: an ambient training scheme to mask out loss from misaligned areas, and a global consistency loss leveraging MS-SWD He et al. (2024) to ensure structural similarity between the output and the ground truth.

Figure 2: Overview of *ImpResDescan*. **Red panel** shows the implicit color correction module with a global feature encoder $\varepsilon_\phi$ that predicts the scale-shift parameters $(\gamma, \beta)$ and produces the color-corrected image $I_c$ from the input scan image $I_s$. **Green panel** illustrates training of the degradation-aware refinement module with a keypoint-based spatial matching. **Purple panel** shows inference of the degradation-aware refinement module.

## 3 PRELIMINARIES

### 3.1 RESFUSION: RESIDUAL NOISE DIFFUSION FOR IMAGE RESTORATION

Resfusion Shi et al. (2024) accelerates diffusion-based restoration by injecting a residual $R = \hat{x}_0 - x_0$ into the forward process and starting the reverse process from the noisy degraded input $\hat{x}_0$ (rather than pure noise), exploiting the low-frequency content preserved in degradations. The forward transition is

$$q(x_t \mid x_{t-1}, R) = \mathcal{N}\big(x_t; \sqrt{\alpha_t}\, x_{t-1} + (1 - \sqrt{\alpha_t})\, R, \, (1 - \alpha_t)I\big), \tag{1}$$

with a standard DDPM schedule $\{\alpha_t\}$. This yields a *residual-noise* target (a weighted mix of Gaussian noise and $R$) predicted at each step. The reverse transition is

$$p_\Omega(x_{t-1} \mid x_t) = \mathcal{N}\big(x_{t-1}; \mu_\Omega(x_t, t), \Sigma_\Omega(x_t, t)\big), \tag{2}$$

where $(\mu_\Omega, \Sigma_\Omega)$ are predicted by a network conditioned on $(x_t, \hat{x}_0)$. Thus, Resfusion achieves strong results across diverse restoration tasks with fewer sampling steps, while using a standard U-Net and off-the-shelf noise schedules.

### 3.2 AMBIENT DIFFUSION: LEARNING FROM CORRUPTED DATA

Ambient Diffusion Daras et al. (2023) extends diffusion models to settings where only corrupted observations are available for training a common case when collecting clean data is infeasible, costly, or raises privacy concerns. Instead of observing clean samples $x_0$, the learner receives $y_0 = Ax_0$; during training, an auxiliary corruption $\tilde{A}$ is also sampled to form $\tilde{y}_0 = \tilde{A}x_0$. The forward noising process is $x_t = x_0 + \sigma_t\eta$, with $\eta \sim \mathcal{N}(0, I)$. The network $h_\Omega$ is trained to approximate $\mathbb{E}[x_0 \mid \tilde{A}x_t, \tilde{A}]$ by minimizing

$$J_{\text{corr}}(\Omega) = \tfrac{1}{2}\, \mathbb{E}_{x_0, t, A, \tilde{A}} \Big\| A\big(h_\Omega(\tilde{A}, \tilde{A}x_t, t) - x_0\big) \Big\|_2^2, \quad \text{where } y_0 = Ax_0, \; \tilde{y}_0 = \tilde{A}x_0, \; x_t = x_0 + \sigma_t\eta. \tag{3}$$

At inference, the model estimates a clean image from only *partial or corrupted measurements* (e.g., masked or projected observations), while avoiding simple memorization of the training data.

## 4 METHODS

Figure 2 summarizes *ImpResDescan*: an implicit correction module uses a global encoder to predict scale–shift $(\gamma, \beta)$ for transforming scanned image $I_s \rightarrow$ color-corrected image $I_c$, followed by a degradation-aware refinement trained with keypoint-based spatial matching and applied at inference to produce the descanned image.

### 4.1 IMPLICIT COLOR CORRECTION

The implicit color correction module is designed to learn a direct mapping from a scanned image $I_s$ to a color-corrected image $I_c$, enabling robust removal of complex color distortions. The architecture comprises a global feature encoder and a pixel-wise correction network. Given a scanned RGB image $I_s$, the global feature encoder $\mathcal{E}_\phi$ extracts a latent vector $\mathbf{g} = \mathcal{E}_\phi(I_s)$, which is used together with each pixel value $I_s(x)$ as input to a multilayer perceptron $\mathcal{F}_\theta$, producing the corrected pixel $I_c(x) = \mathcal{F}_\theta(I_s(x), \mathbf{g})$. The parameters $\phi$ and $\theta$ are optimized jointly by minimizing the mean squared error between $I_c$ and the clean target image $I_o$:

$$\mathcal{L}(\theta, \phi) = \frac{1}{N} \sum_x \|I_c(x) - I_o(x)\|^2, \tag{4}$$

where $N$ is the total number of pixels. In practice, this implicit formulation proves effective, as the network learns to adapt its corrections to scanner-specific distortions directly from data, yielding more consistent and accurate restorations across devices. Moreover, the encoder implicitly captures scanner-specific error responses and can cluster different scanners according to their error characteristics. Further details of this analysis are provided in the supplementary material, Section A5.1.

### 4.2 DEGRADATION-AWARE REFINEMENT STAGE

The second stage of our pipeline is a **degradation-aware local refinement module** that targets local distortions such as noise, halftone patterns, and bleed-through effects. This stage uses a residual diffusion network with a U-Net backbone, conditioned on a distortion encoder that estimates the degradation in each input. Robustness to misaligned data pairs is achieved by employing an ambient training strategy with spatial masking during training.

#### 4.2.1 DISTORTION AWARE ENCODER

The distortion-aware encoder is trained to estimate the level of distortion present in each image. For supervision, we synthetically generate mixtures by randomly combining patches from the color corrected image $I_c$ and the original image $I_o$, yielding a composite input $I_m$ and an associated *distortion scale* $s \in [0, 1]$, where $s = 0$ denotes a fully original image, $s = 1$ denotes a color-corrected image, and intermediate values represent partial mixtures.

Formally, for given $I_c$, $I_o$, and distortion scale $s$:

$$I_m = \mathrm{Mix}(I_c, I_o; s), \tag{5}$$

where $\mathrm{Mix}(\cdot)$ denotes the patch-based mixing process. The encoder $\mathcal{E}_\psi$ maps $I_m$ to an embedding:

$$\mathbf{e}_m = \mathcal{E}_\psi(I_m). \tag{6}$$

To ensure a meaningful embedding space, we employ an *ordinal contrastive loss* that aligns embedding distances with differences in distortion scale. For a batch of embeddings $\{\mathbf{e}_i\}$ and scales $\{s_i\}$, for each pair $(i, j)$, we define

$$d_{ij} = \|\mathbf{e}_i - \mathbf{e}_j\|_2, \ t_{ij} = m |s_i - s_j|. \tag{7}$$

where $m$ is a margin hyperparameter. The loss penalizes deviations from the target embedding distance:

$$\mathcal{L}_{\mathrm{dist}}(\psi) = \frac{1}{B(B-1)} \sum_{i \neq j} |s_i - s_j| \left[ \mathbb{I}(d_{ij} > t_{ij})(d_{ij} - t_{ij})^2 + \mathbb{I}(d_{ij} < t_{ij})(t_{ij} - d_{ij})^2 \right], \tag{8}$$

where $\mathbb{I}(\cdot)$ is the indicator function and $B$ is the batch size. This loss ensures the embedding space is both order- and scale-aware, with similar distortion levels mapped closely and larger differences mapped farther apart. As the encoder is trained to capture the distortion scale, it can also be applied to color-corrected images to provide effective supervision for non-color degradations.

### 4.2.2 KEYPOINT-BASED SPATIAL MASKING

To address misalignment between the color-corrected image $I_c$ and the original reference $I_o$, we construct a spatial mask $\mathcal{M}$ based on dense keypoint correspondences obtained via GIM Shen et al. (2024b). Only keypoint pairs with spatial distance below a predefined threshold are retained. For each valid correspondence, we activate a local square patch in both images; the final mask is defined as the intersection of these patches:

$$\mathcal{M} = \mathcal{M}_c \wedge \mathcal{M}_o, \tag{9}$$

where $\mathcal{M}_c$ and $\mathcal{M}_o$ are binary masks and $\wedge$ denotes element-wise AND.

This mask is incorporated into the Resfusion loss:

$$\mathcal{L}_{\text{ambient}}(\Omega) = \mathbb{E}_{I_o, I_c, t, \epsilon} \left[ \frac{\left\| \mathcal{M} \odot \left( \text{res}\,\epsilon - \widehat{\text{res}\,\epsilon} \right) \right\|_2^2}{\sum_{i,j} \mathcal{M}_{i,j}} \right], \tag{10}$$

$$\text{where} \qquad \text{res}\,\epsilon = \epsilon + \frac{1 - \sqrt{\alpha_t}}{\sqrt{1 - \alpha_t}} \frac{1}{\beta_t}(I_c - I_o), \qquad \widehat{\text{res}\,\epsilon} = \text{res}\,\epsilon_\Omega(x_t, I_c, \mathbf{e}_c, t).$$

By restricting supervision to reliably aligned regions, this approach ensures robust training under imperfect registration and improves restoration quality in real-world scanned documents.

### 4.2.3 GLOBAL CONSISTENCY LOSS

To capture global image structure beyond the aligned regions used in supervised training, we introduce a multi-scale Sliced Wasserstein Distance (MS-SWD) loss between the predicted clean image and a global reference. Unlike pixel-wise losses, MS-SWD encourages overall visual consistency and does not require spatial alignment.

At each diffusion step $t$, we estimate the clean image as

$$\hat{x}_0 = \frac{x_t - \frac{\beta_t}{\sqrt{1 - \bar{\alpha}_t}} \widehat{res\epsilon}}{\sqrt{\bar{\alpha}_t}}, \tag{11}$$

where $\alpha_t$ and $\beta_t$ are schedule parameters. The global consistency loss is defined as

$$\mathcal{L}_{\text{global}}(\Omega) = \text{MS-SWD}(\hat{x}_0, I_o). \tag{12}$$

where $I_o$ is the reference image. This loss promotes globally coherent outputs and regularizes the diffusion process.

**Overall Objective.** The overall training objective combines the ambient alignment loss and the global loss:

$$\mathcal{L}_{\text{total}} = \mathcal{L}_{\text{ambient}} + \lambda \mathcal{L}_{\text{global}}, \tag{13}$$

where $\mathcal{L}_{\text{ambient}}$ denotes the masked alignment loss and $\mathcal{L}_{\text{global}}$ is the global consistency loss computed via MS-SWD. $\lambda$ is a balancing weight that controls the trade-off between local alignment fidelity and global structural consistency, and is fixed to $0.1$ in all experiments.

**Baselines and Evaluation.** As descanning remains an underexplored task, rigorous benchmarking is challenging due to the scarcity of established baselines; to date, only one method DescannDiffusion Cha et al. (2024) directly targets this problem. To ensure a thorough and transparent evaluation, we compare our approach against methods from three relevant categories: (i) **Commercial products**, including Clear Scan IndyMobileApp (2016), Adobe Scan Adobe (2017), and Microsoft Lens Microsoft (2015); (ii) **Image restoration methods**, including image-to-image translation networks (Pix2PixHD Wang et al. (2018), CycleGAN Zhu et al. (2017)), advanced restoration models (HDRUNet Chen et al. (2021), Restormer Zamir et al. (2022), ESDNet Yu et al. (2022), NAFNet Chen et al. (2022)), real-world photo restoration methods (OPR Wan et al. (2020), DPS Ho & Zhou (2022)), latent diffusion-based framework Yue et al. (2025)) and document image restoration method Zhang et al. (2024); and (iii) **Descanning methods**, with only DescannDiffusion serving as a direct baseline. While our evaluation is as comprehensive as possible, this limitation highlights both the novelty of our work and the current gap in the field.

**Algorithm 1** Training of Degradation-Aware Refinement

**Require:** $(I_s, I_o)$, model res $\epsilon_\Omega$, **pre-trained** ColorCorrect, **pre-trained** distortion aware encoder $\mathcal{E}_\psi$, timestep $T$
1: $I_c \leftarrow \text{ColorCorrect}(I_s)$
2: $\mathbf{e}_c \leftarrow \mathcal{E}_\psi(I_c)$
3: Compute mask $\mathcal{M}$ for $(I_c, I_o)$
4: $R \leftarrow I_c - I_o$
5: $t \sim \mathcal{U}\{1, T\}, \quad \epsilon \sim \mathcal{N}(0, I)$
6: $x_t \leftarrow \sqrt{\bar{\alpha}_t} I_o + (1 - \sqrt{\bar{\alpha}_t})R + \sqrt{1 - \bar{\alpha}_t}\,\epsilon$
7: $res\epsilon \leftarrow \epsilon + \frac{1 - \sqrt{\alpha_t}}{\sqrt{1 - \alpha_t}\,\beta_t} R$
8: $\widehat{res}\epsilon \leftarrow \text{res}\,\epsilon_\Omega(x_t, I_c, \mathbf{e}_c, t)$
9: $\mathcal{L}_{\text{ambient}} \leftarrow \frac{\|\mathcal{M} \odot (res\epsilon - \widehat{res}\epsilon)\|_2^2}{\sum_{i,j} \mathcal{M}_{i,j}}$
10: $\hat{x}_0 \leftarrow \frac{x_t - \frac{\beta_t}{\sqrt{1 - \bar{\alpha}_t}} \widehat{res}\epsilon}{\sqrt{\bar{\alpha}_t}}$
11: $\mathcal{L}_{\text{global}} \leftarrow \text{MS-SWD}(\hat{x}_0, I_o)$
12: $\mathcal{L}_{\text{total}} \leftarrow \mathcal{L}_{\text{ambient}} + \lambda\,\mathcal{L}_{\text{global}}$
13: Update **only** res $\epsilon_\Omega$ to minimize $\mathcal{L}_{\text{total}}$

**Algorithm 2** Inference of Degradation-Aware Refinement

**Require:** $I_s$ (scanned image), **pre-trained** ColorCorrect, **pre-trained** $\mathcal{E}_\psi$, **trained** res $\epsilon_\Omega$, $\{\beta_t\}$, timestep $T'$
1: $I_c \leftarrow \text{ColorCorrect}(I_s)$
2: $\mathbf{e}_c \leftarrow \mathcal{E}_\psi(I_c)$
3: $\epsilon \sim \mathcal{N}(0, I)$
4: $x_{T'} \leftarrow (1 - \sqrt{\alpha_{T'}})I_c + \sqrt{1 - \alpha_{T'}}\,\epsilon$
5: **for** $t = T', \dots, 1$ **do**
6: $\quad \widehat{res}\epsilon \leftarrow \text{res}\,\epsilon_\Omega(x_t, I_c, \mathbf{e}_c, t)$
7: $\quad \mu \leftarrow \frac{1}{\sqrt{\alpha_t}}\left(x_t - \frac{\beta_t}{\sqrt{1 - \alpha_t}} \widehat{res}\epsilon\right)$
8: $\quad z \sim \mathcal{N}(0, I)$ if $t > 1$ else $z = 0$
9: $\quad x_{t-1} \leftarrow \mu + \sqrt{\beta_t}\,z$
10: **end for**
11: **return** $x_0$

| Method | PSNR (dB) ↑ | SSIM ↑ | LPIPS ↓ | FID ↓ |
|---|---|---|---|---|
| Pix2PixHD | 20.58 | 0.8014 | 0.057 | 18.30 |
| CycleGAN | 21.52 | 0.8417 | 0.050 | 16.99 |
| HDRUNet | 20.90 | 0.8480 | 0.055 | 16.42 |
| Restormer | 20.37 | 0.7915 | 0.152 | 25.57 |
| ESDNet | 21.22 | 0.8418 | 0.088 | 15.24 |
| NAFNet | 22.03 | 0.8538 | 0.048 | 16.00 |
| OPR* | 18.09 | 0.7249 | 0.158 | 21.45 |
| DPS* | 17.93 | 0.7354 | 0.150 | 41.64 |
| ResShift | 21.80 | 0.8359 | 0.066 | 15.14 |
| DocRes | 23.89 | 0.8924 | 0.056 | 15.52 |
| Clear Scan | 21.46 | 0.8183 | 0.054 | 18.09 |
| Adobe Scan | 15.80 | 0.6153 | 0.141 | 23.55 |
| Microsoft Lens | 20.48 | 0.8013 | 0.056 | 18.97 |
| DescanDiffusion+ | 23.23 | 0.8863 | 0.059 | 15.40 |
| ImpResDescan | **24.20** | **0.8996** | **0.048** | **11.97** |

Table 1: Quantitative comparison of descanning performance on original DESCAN-18K test set (average PSNR/SSIM/LPIPS/FID). Methods with an asterisk(*) are pre-trained versions.

### 4.3 COMPARISON TO EXISTING METHODS

We employ four quantitative metrics to evaluate descanning performance: PSNR, which measures pixel-wise fidelity between the restored and original image; SSIM Wang et al. (2004) and LPIPS Zhang et al. (2018), which assess perceptual quality; and Fréchet Inception Distance (FID) Heusel et al. (2017), which provides an overall measure of generation performance.

Quantitative results are presented in Table 1. ImpResDescan outperforms all competing approaches including commercial products across all reported metrics. In particular, it achieves state-of-the-art scores on PSNR ↑, SSIM ↑, LPIPS ↓, and FID ↓, indicating both high perceptual realism and strong fidelity in the restored images.

Further evaluation, summarized in Table 2, employs alignment-robust full-reference metrics to assess robustness to spatial misalignment. CW-SSIM Wang & Simoncelli (2005) captures phase-consistent structural similarity and tolerates small shifts; DISTS Ding et al. (2020) measures perceptual distance in deep features reflecting structural and textural fidelity; VSI Zhang et al. (2014) is

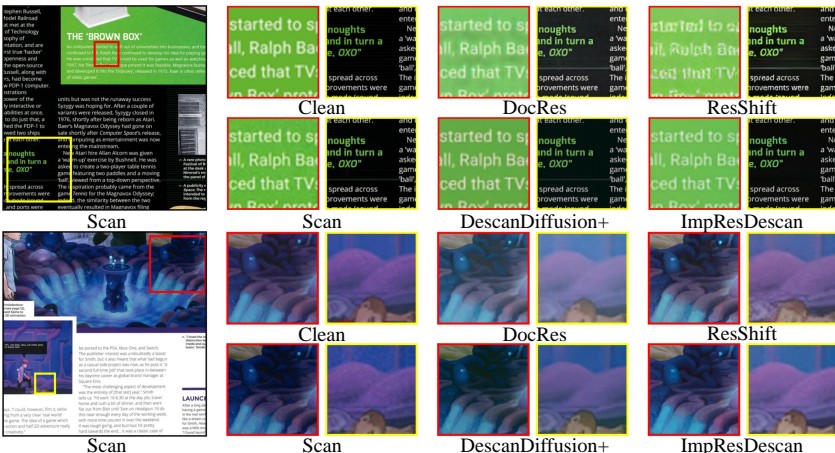

Figure 3: Visual comparison of descanning results on the DESCAN-18K test set. The input scans present a combination of degradations, including color shifts, texture distortions, and noise. The task requires simultaneous correction of these multiple distortion types. Our method effectively restores both color accuracy and structural integrity, resulting in outputs that closely resemble the original images.

| Method | Full-reference metrics | | | | Non-reference metrics | | | |
|---|---|---|---|---|---|---|---|---|
| | CW-SSIM ↑ | DISTS ↓ | VSI ↑ | AHIQ ↑ | LIQE ↑ | PAQ2PIQ ↑ | MUSIQ ↑ | Q-Align ↑ |
| DescanDiffusion+ | 0.9777 | 0.0804 | 0.9815 | 0.4935 | 3.7898 | 70.5887 | 62.0146 | 3.6641 |
| DocRes | 0.9783 | 0.0889 | 0.9826 | 0.5464 | 3.7213 | 70.5597 | 61.3005 | 3.4531 |
| ImpResDescan | **0.9818** | **0.0688** | **0.9831** | **0.5500** | **3.9879** | **70.9170** | **62.4829** | **3.6797** |

Table 2: Experiments on alignment-robust full-reference metrics and non-reference metrics.

a saliency-weighted fidelity index emphasizing perceptually important regions; and AHIQ Lao et al. (2022) is an attention-based, artifact-aware full-reference IQA. ImpResDescan ranks first across all four metrics in Table 2, indicating stronger structure preservation, fewer perceptual deviations, and reduced artifacts compared with DescanDiffusion+ and DocRes.

We also evaluate non-reference IQA metrics (NR-IQA) that do not require a ground-truth reference. LIQE Zhang et al. (2023) measures naturalness under learned image priors; PAQ2PIQ Ying et al. (2020) predicts a global quality score informed by the severity of artifacts (e.g., blockiness, oversharpening, residual noise); MUSIQ Ke et al. (2021) estimates scene-aware perceptual quality from multi-scale features; and Q-Align Wu et al. (2023) provides human-aligned ratings via a large multimodal model. ImpResDescan delivers consistent gains across these non-reference metrics, indicating higher overall perceptual quality and reduced artifact prominence without a reference. Additional detailed analysis can be found in the supplementary material, Section A4.

Fig. 3 presents visual results from several methods. The baseline Descandiffusion+ fails to adequately restore both color and other degradations in scanned images. In contrast, ImpResDescan effectively recovers degradations and produces outputs that closely resemble the original images. Latent diffusion-based approaches such as ResShift generate perceptually pleasing results but struggle to reconstruct finer details. Additional visual results can be found in Section A6.

## 4.4 ABLATION STUDY

We conduct a progressive ablation (Table 3) to isolate each module's effect. Adding *ambient training* (col. (b)) improves all metrics over (a) and yields the highest SSIM. Introducing *color correction* (col. (c)) further increases PSNR and AHIQ, with metrics otherwise comparable to (b). Incorporating the *distortion encoder* (col. (d)) slightly raises PSNR and improves LPIPS and FID, but reduces SSIM and AHIQ relative to (c). Finally, adding the *global loss* (col. (e)) delivers the strongest overall

|  | (a) | (b) | (c) | (d) | (e) |
|---|---|---|---|---|---|
| Ambient training | ✗ | ✓ | ✓ | ✓ | ✓ |
| Color correction | ✗ | ✗ | ✓ | ✓ | ✓ |
| Distortion encoder | ✗ | ✗ | ✗ | ✓ | ✓ |
| Global loss | ✗ | ✗ | ✗ | ✗ | ✓ |
| PSNR (dB) ↑ | 23.52 | 23.70 | 24.09 | 24.1489 | **24.20** |
| SSIM ↑ | 0.8902 | **0.9007** | 0.8958 | 0.8891 | 0.8996 |
| LPIPS ↓ | 0.0556 | 0.0518 | 0.0530 | 0.0496 | **0.0486** |
| FID ↓ | 13.69 | 13.13 | 13.24 | 12.50 | **11.97** |
| AHIQ ↑ | 0.5194 | 0.5408 | 0.5473 | 0.5454 | **0.5500** |

Table 3: Progressive ablation of each component.

| Method | Inference Time |
|---|---|
| CycleGAN | $10^{-5}$s |
| Restormer | 0.5289s |
| ESDNet | 0.2251s |
| NAFNet | 0.0013s |
| DescanDiffusion+ | 2.5827s |
| ImpResDescan | 1.2610s |

Table 4: Inference time comparison.

performance best PSNR ↑, LPIPS ↓, FID ↓, and AHIQ ↑ while maintaining SSIM ↑ close to the peak at (b). These trends indicate complementary contributions from all components and the benefit of the full configuration. Additional ablation studies are provided in Section A3 of the supplementary material.

| Dataset | Method | LIQE ↑ | PAQ2PIQ ↑ | MUSIQ ↑ | Q-ALIGN ↑ |
|---|---|---|---|---|---|
| Comic | DescanDiffusion+ | 3.0728 | 75.8179 | 65.7674 | 3.6833 |
| | DocRes | 3.0365 | 76.7937 | 66.0620 | 3.6413 |
| | ImpResDescan | **3.2064** | **77.3089** | **66.7808** | **3.7224** |
| DPS | DescanDiffusion+ | 2.5107 | 71.8125 | 58.4503 | 2.8213 |
| | DocRes | 2.0202 | 69.7195 | 55.6682 | 2.5388 |
| | ImpResDescan | **2.6520** | **72.9666** | **58.5963** | **2.8386** |

Table 5: Experiments on external datasets (Comic and DPS) evaluated with NR-IQA.

| Method | WER ↓ | NED ↓ |
|---|---|---|
| ResShift | 0.7602 | 0.4523 |
| DescanDiffusion+ | 0.5664 | 0.2657 |
| DocRes | 0.5534 | 0.2564 |
| ImpResDescan | **0.5472** | **0.2517** |

Table 6: Experiments on the OCR task using DESCAN-18K.

## 4.5 GENERALIZATION TO EXTERNAL DATASETS AND OCR ROBUSTNESS

We evaluate our method on two external datasets: Digital Comic Museum (Comic) D. C. Museum (2022) and DPS Ho & Zhou (2022). The Comic dataset contains 89 pages of scanned comics, while DPS consists of 100 smartphone-scanned images. As ground-truth references are unavailable, we report four non-reference IQA (NR-IQA) metrics: LIQE Zhang et al. (2023), PAQ2PIQ Ying et al. (2020), MUSIQ Ke et al. (2021), and Q-Align Wu et al. (2023). As shown in Table 5, ImpResDescan consistently outperforms the Descan baseline across all metrics on both datasets.

We further evaluate OCR Jaided (2020) results on DESCAN-18K using Word Error Rate (WER) Lcvenshtcin (1966) and Normalized Edit Distance (NED) Marzal & Vidal (2002). As shown in Table 6, ImpResDescan achieves the lowest errors, demonstrating improved text recognition compared to other methods despite not being explicitly designed for text enhancement.

## 4.6 INFERENCE TIME EVALUATION

Table 4 summarizes the inference times of several models on an NVIDIA TESLA V100 GPU. ImpResDescan employs a residual diffusion approach, enabling image reconstruction from the scanned input in just five steps. This efficient design significantly reduces inference time compared to other diffusion-based methods, while maintaining strong restoration quality.

## 5 CONCLUSION

We introduced **ImpResDescan**, a robust diffusion-based framework for restoring scanned images degraded by complex, real-world artifacts, including nonlinear color shifts, local degradations, and spatial misalignments. ImpResDescan is trained using the proposed ambient training strategy and is regularized with a global consistency loss, which together ensure robustness to spatial misalignment and enable effective restoration from unaligned, real-world data. By leveraging these components, our method learns to recover both global structure and fine-grained local details, even in challenging, misaligned scenarios. Extensive experiments on the DESCAN-18K dataset demonstrate that ImpResDescan sets a new benchmark for scan restoration, paving the way for more reliable document enhancement and robust image restoration in practical settings.

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

# A APPENDIX

## LLM USAGE

We used an AI-based assistant (ChatGPT) solely for minor language editing and polishing. All research ideas, experimental design, and analyses were conducted by the authors.

## A1 IMPLEMENTATION DETAILS

**Implicit Color Correction.** The color encoder $\mathcal{E}_\phi$ is implemented as a ResNet-34 backbone He et al. (2016) truncated before the classification head. A global average pooling layer produces a 512-dimensional feature vector:

$$\mathbf{g} = \mathcal{E}_\phi(I_s) \in \mathbb{R}^{512}, \tag{14}$$

where $I_s \in \mathbb{R}^{H \times W \times 3}$ denotes the scanned RGB image.

The pixel-wise correction network $\mathcal{F}_\theta$ is a four-layer MLP with hidden width 256. For each pixel $x$, the observed value $I_s(x) \in \mathbb{R}^3$ is mapped to a hidden representation:

$$h^{(0)}(x) = \sigma(W^{(0)}I_s(x) + b^{(0)}), \tag{15}$$

where $W^{(0)} \in \mathbb{R}^{256 \times 3}$, $b^{(0)} \in \mathbb{R}^{256}$, and $\sigma(\cdot)$ is the SiLU activation Elfwing et al. (2018).

From the global feature $\mathbf{g}$, the network predicts per-layer modulation parameters:

$$\{\gamma^{(l)}(\mathbf{g}), \ \beta^{(l)}(\mathbf{g})\} = W^{(l)}_{\text{mod}}\mathbf{g} + b^{(l)}_{\text{mod}}, \quad l = 1, \ldots, L-2, \tag{16}$$

where $\gamma^{(l)}$ and $\beta^{(l)}$ represent learned scale and shift vectors, respectively.

Each hidden layer is then modulated as:

$$h^{(l)}(x) = \sigma\Big(\gamma^{(l)}(\mathbf{g}) \odot h^{(l-1)}(x) + \beta^{(l)}(\mathbf{g})\Big), \tag{17}$$

with $\odot$ denoting element-wise multiplication.

Finally, the corrected pixel value is obtained by mapping the last hidden representation to RGB:

$$I_c(x) = W^{(L)}h^{(L-1)}(x) + b^{(L)}, \quad I_c(x) \in \mathbb{R}^3. \tag{18}$$

This formulation enables the global feature $\mathbf{g}$ to adaptively control the scale and bias of pixel-level activations, allowing the model to learn scanner-specific corrections in a spatially consistent way.

**Distortion-Aware Encoder** The distortion-aware encoder $\mathcal{E}_\psi$ is designed to capture the severity of distortions in the images. Its supervision is obtained by constructing synthetic mixtures $I_m$ from the color-corrected image $I_c$ and the ground-truth original $I_o$, where patch- or region-level replacements simulate different distortion scales $s \in [0, 1]$.

The encoder is implemented using a ResNet-34 backbone He et al. (2016) truncated before the classification head, followed by a linear projection to a low-dimensional embedding space of size $d = 128$. Formally,

$$\mathbf{e}_m = \mathcal{E}_\psi(I_m), \qquad \mathbf{e}_m \in \mathbb{R}^d, \tag{19}$$

where $\mathbf{e}_m$ is L2-normalized to enforce $\|\mathbf{e}_m\|_2 = 1$. This embedding provides a compact representation of the distortion characteristics, enabling downstream modules to adaptively modulate restoration according to the input's degradation profile.

**Diffusion Backbone with Distortion-Aware Conditioning** The denoising network adopts a U-Net architecture conditioned on the distortion embedding, following the residual diffusion framework of RDDM Liu et al. (2024). Specifically, the distortion embedding of $I_c$, denoted $\mathbf{e}_c \in \mathbb{R}^d$ is injected into every residual block through cross-attention modules, allowing the model to adaptively attend to degradation characteristics.

The U-Net consists of four downsampling and four upsampling stages. Each stage contains two ResNet-style blocks He et al. (2016) of the form

$$h_{l+1} = \text{Block}(h_l, \gamma(t), \mathbf{e}_c) + h_l, \tag{20}$$

where $h_l$ denotes the feature map at layer $l$, $\gamma(t)$ is the sinusoidal positional time embedding Vaswani et al. (2017) projected through an MLP, and $\mathbf{e}_c$ provides the conditioning via cross-attention. Downsampling is implemented with stride-2 convolutions, while upsampling uses nearest-neighbor interpolation followed by $3 \times 3$ convolution.

At the bottleneck, two residual blocks are interleaved with a cross-attention layer:

$$h' = \text{CrossAttn}(h, \mathbf{e}_c), \tag{21}$$

which aligns intermediate features with the distortion embedding.

The base filter width is set to 64 in the first downsampling block and doubles after each stage, yielding channel sizes of $\{64, 128, 256, 512\}$. SiLU activation Elfwing et al. (2018) is used throughout.

## A2 TRAINING AND INFERENCE OF IMPRESDESCAN

**Implicit Color Correction Module** The color encoder and modulation MLP are jointly trained on the DESCAN-18K training set. During each iteration, paired scan and clean images are resized to $512 \times 512$ pixels. The scan image is also resized to $256 \times 256$ and passed through a ResNet-34 color encoder to obtain a global feature vector. This feature vector is used to generate per-layer scale-and-shift conditioning parameters for the MLP, which predicts the pixelwise color-corrected output. The entire model is optimized end-to-end using mean squared error loss between the predicted and clean images, with a batch size of 64 and the Adam optimizer.

**Distortion-Aware Encoder** The distortion-aware encoder is pre-trained to capture degradation characteristics in scan images. Training samples are generated by mixing clean and scanned patches at varying proportions, using a patch size of $64 \times 64$. The scan-to-clean mixing ratios are sampled from a uniform distribution over $[0, 1]$, allowing the network to observe a continuous range of degradation severities. Each mixed patch is assigned a distortion score equal to its scan content ratio. The encoder is trained using an ordinal contrastive loss, which encourages the embedding distance to reflect differences in distortion severity: patches with similar scores are embedded nearby, while those with dissimilar scores are pushed farther apart. Pre-training is performed on the DESCAN-18K dataset for 1000 epochs, using a batch size of 1024, a learning rate of $1 \times 10^{-3}$, and the Adam optimizer.

**Degradation-Aware Refinement** This stage is trained following Algorithm 1 in the main manuscript. We use the pretrained color encoder and distortion encoder. Keypoint-based spatial masks are generated for the DESCAN-18K dataset to provide robust supervision under imperfect registration; dense local correspondences are established between each color-corrected scan and its digital reference using the GIM Shen et al. (2024a) generalizable image matcher.

The denoising network is trained for $T = 12$ time steps with a linear noise schedule. Optimization is performed using the AdamW optimizer with cosine annealing for the learning rate schedule. We use a base learning rate of $8.8 \times 10^{-4}$, a minimum learning rate of $3 \times 10^{-5}$, and set weight decay to 0. The loss trade-off parameter $\lambda$ is set to 0.1 throughout training. The model is trained for 576,000 iterations on a cluster of 8 NVIDIA RTX 4090 GPUs with a batch size of 16.

**Inference** During inference, we follow Algorithm 2 in the main manuscript. The reverse diffusion process is initialized from the color-corrected image $I_c$ at an intermediate time step $T'$, rather than from pure Gaussian noise. The value of $T'$ is selected such that $\sqrt{\bar{\alpha}_{T'}} \approx 0.5$ under the linear noise schedule, which corresponds to $T' = 5$ for $T = 12$ steps in our setting. The diffusion model then iteratively denoises $I_c$ for $T'$ steps to produce the restored image, efficiently utilizing information from the observed input for improved restoration quality.

## A3    ABLATION OF KEYPOINT-BASED AMBIENT TRAINING

**Patch size and thresholding**    We conduct an ablation study on the keypoint-based ambient training framework by training our end-to-end model with varying patch sizes and keypoint distance thresholds used for spatial masking. All experiments strictly follow the training methodology outlined in Algorithm 1 of the main paper to ensure consistency. We evaluate the impact of spatial supervision configurations on model performance by analyzing variations in reconstruction fidelity, structural similarity, perceptual quality, and distributional alignment. The results reported below reflect model performance on the test set of the DESCAN-18K dataset.

| Patch Size | Threshold | PSNR (dB) ↑ | SSIM ↑ | LPIPS ↓ | FID ↓ |
|---|---|---|---|---|---|
| 8 | 0.1 | 22.2493 | 0.8747 | 0.0730 | 21.4038 |
| | 0.5 | 24.1200 | 0.8947 | 0.0501 | 13.3250 |
| | 1.0 | 23.6140 | 0.8934 | 0.0529 | 13.2935 |
| 16 | 0.1 | 24.0462 | 0.8988 | 0.0519 | 13.5405 |
| | 0.5 | 23.6980 | 0.8922 | 0.0540 | 13.4673 |
| | 1.0 | 24.1340 | 0.8947 | 0.0494 | 12.6575 |
| 64 | 0.1 | 24.0701 | 0.8966 | 0.0513 | 12.7864 |
| | 0.5 | 24.1983 | 0.8996 | 0.0486 | 11.9779 |
| | 1.0 | 23.9528 | 0.8954 | 0.0513 | 13.0173 |

Table 7: Ablation study of Patch Size and Distance Threshold. Red and blue indicate 1$^{st}$ and 2$^{nd}$ best values across all settings for each metric, respectively.

| Method | Full-reference metrics | | | |
|---|---|---|---|---|
| | PSNR (dB) ↑ | SSIM ↑ | LPIPS ↓ | FID ↓ |
| Soft Thresholding | 23.7962 | 0.8926 | 0.0535 | 13.3164 |
| Binary Masking | **24.1983** | **0.8996** | **0.0486** | **11.9779** |

Table 8: Comparison of Soft Mask and Binary Masking.

**Soft-mask training vs. binary masking.**    We evaluate the full pipeline using a *soft* mask for the ambient training stage instead of a binary mask, while keeping all other components the same (color correction, distortion encoder, global loss). The soft mask is built by applying a Gaussian around each clean/scan keypoint to form maps $A$ (clean) and $B$ (scan), and then combining the two:

$$S = \alpha A + \beta B, \qquad \sigma = 10 \text{ px}, \ \alpha = \beta = 1.$$

As shown in Table 8, the binary mask performs better across all reported metrics in our setup.

| Method | Scanner | CW-SSIM ↑ | LPIPS ↓ | DISTS ↓ | FID ↓ |
|---|---|---|---|---|---|
| DescanDiffusion+ | Fuji Xerox ApeosPort C2060 | 0.9827 | 0.0565 | 0.0782 | 16.3813 |
| | Canon imagePRESS C650 | 0.9726 | 0.0632 | 0.0826 | 19.0941 |
| | Canon imageRUNNER ADVANCE 6265 | 0.9682 | 0.0666 | 0.0830 | 19.6140 |
| | Plustek OpticBook 4800 | 0.9661 | 0.0736 | 0.0980 | 28.7039 |
| DocRes | Fuji Xerox ApeosPort C2060 | 0.9796 | 0.0577 | 0.0918 | 17.7126 |
| | Canon imagePRESS C650 | 0.9771 | 0.0543 | 0.0859 | 18.0314 |
| | Canon imageRUNNER ADVANCE 6265 | 0.9764 | 0.0499 | 0.0988 | 17.9299 |
| | Plustek OpticBook 4800 | 0.9805 | 0.0427 | 0.0843 | 23.3719 |
| ImpResDescan | Fuji Xerox ApeosPort C2060 | 0.9832 | 0.0506 | 0.0715 | 13.8530 |
| | Canon imagePRESS C650 | 0.9804 | 0.0467 | 0.0660 | 13.7716 |
| | Canon imageRUNNER ADVANCE 6265 | 0.9808 | 0.0461 | 0.0604 | 13.0886 |
| | Plustek OpticBook 4800 | 0.9830 | 0.0402 | 0.0573 | 18.0773 |

Table 9: CW-SSIM, LPIPS, DISTS, and FID results for full-reference image quality evaluation per scanner.

| Method | Scanner | LIQE ↑ | PAQ2PIQ ↑ | MUSIQ ↑ | Q-ALIGN ↑ |
|---|---|---|---|---|---|
| DescanDiffusion+ | Fuji Xerox ApeosPort C2060 | 3.7404 | 70.6380 | 62.1136 | 3.6777 |
| | Canon imagePRESS C650 | 3.8402 | 70.5384 | 61.9155 | 3.6504 |
| | Canon imageRUNNER ADVANCE 6265 | 4.1119 | 71.4662 | 64.8746 | 3.6602 |
| | Plustek OpticBook 4800 | 3.2961 | 69.9248 | 58.4694 | 3.3477 |
| DocRes | Fuji Xerox ApeosPort C2060 | 3.6800 | 70.6626 | 61.4670 | 3.4746 |
| | Canon imagePRESS C650 | 3.7620 | 70.4558 | 61.1339 | 3.4297 |
| | Canon imageRUNNER ADVANCE 6265 | 4.1124 | 71.0783 | 64.6979 | 3.5176 |
| | Plustek OpticBook 4800 | 3.4627 | 70.0561 | 59.3461 | 3.3730 |
| ImpResDescan | Fuji Xerox ApeosPort C2060 | 3.9118 | 70.9483 | 62.3649 | 3.6836 |
| | Canon imagePRESS C650 | 4.0638 | 70.8847 | 62.6009 | 3.6777 |
| | Canon imageRUNNER ADVANCE 6265 | 4.3635 | 71.3702 | 66.0897 | 3.7480 |
| | Plustek OpticBook 4800 | 3.6686 | 70.3058 | 60.3889 | 3.5254 |

Table 10: Non-reference IQA per scanner: LIQE , PAQ2PIQ, MUSIQ, and Q-ALIGN.

## A4 ADDITIONAL ANALYSIS

To further assess the generalization capability of our models, we evaluate DescanDiffusion+, DocRes, and ImpResDescan on the combined validation and test splits of the DESCAN-18K dataset. This unified evaluation set comprises images scanned from four devices *Plustek OpticBook 4800*, *Canon imageRUNNER ADVANCE 6265*, *Fuji Xerox ApeosPort C2060*, and *Canon image-PRESS C650* and includes only samples that were not used during training.

Table 9 reports standard full-reference image quality metrics CW-SSIM (↑), LPIPS Zhang et al. (2018) (↓), DISTS Ding et al. (2020) (↓), and FID Heusel et al. (2017) (↓) for each scanner individually. Across all devices, ImpResDescan consistently outperforms both DescanDiffusion+ and DocRes, achieving higher CW-SSIM and lower LPIPS, DISTS, and FID, indicating superior structural alignment, perceptual similarity, and distributional consistency with respect to ground-truth images. Results are shown per scanner to provide a complete view of model generalization across heterogeneous acquisition hardware.

Additionally, Table 10 presents non-reference image quality results using LIQE (↑), PAQ2PIQ (↑), MUSIQ (↑), and Q-ALIGN (↑) computed per scanner. As these metrics do not rely on pixel-accurate alignment, they directly assess perceptual quality under scanner-induced shifts. ImpResDescan achieves the best scores across all four scanners *Fuji Xerox ApeosPort C2060*, *Canon imagePRESS C650*, *Canon imageRUNNER ADVANCE 6265*, and *Plustek OpticBook 4800* for LIQE, MUSIQ, and Q-ALIGN, and shows comparable PAQ2PIQ performance to DescanDiffusion+ on *Canon imageRUNNER ADVANCE 6265*. Overall, these findings indicate stronger cross-device perceptual quality and alignment robustness without reference, supporting the generalization of ImpResDescan to previously unseen hardware.

## A5 INTENSITY-DEPENDENT SCANNER ERROR ANALYSIS

Scanner-induced errors vary with image intensity and cannot be effectively corrected using global per-channel adjustments. To analyze this behavior, we compute the mean absolute error (MAE) between scanned images and their corresponding clean digital originals as a function of intensity.

Let $I_o(x, y, c)$ and $I_{\text{scan}}(x, y, c)$ denote the normalized pixel values of the original and scanned images at position $(x, y)$ and channel $c \in \{R, G, B\}$. To reduce the impact of minor spatial misalignments, we apply $4 \times 4$ mean pooling to both images:

$$\tilde{I}_o = \text{Pool}_{4 \times 4}(I_o), \quad \tilde{I}_{\text{scan}} = \text{Pool}_{4 \times 4}(I_{\text{scan}}) \tag{22}$$

The absolute error is then computed on the pooled images as:

$$E(x, y, c) = \left| \tilde{I}_{\text{scan}}(x, y, c) - \tilde{I}_o(x, y, c) \right| \tag{23}$$

Errors are averaged across RGB channels and evaluated at each original intensity level to generate scanner-specific error curves. This analysis is conducted on 720 scanned image pairs from the validation and test sets, covering all four scanners in the dataset.

### A5.1 IMPLICIT COLOR CORRECTION AND SCANNER ERROR ANALYSIS

Our method uses an implicit color correction module trained end-to-end with a per-pixel reconstruction loss. A ResNet-34 encoder He et al. (2016) produces a global feature that modulates a pixel-wise MLP via affine scales and shifts, enabling localized corrections that remain globally consistent across scanners.

Since scanners introduce varying intensity-dependent distortions, we hypothesize that the global encoder implicitly learns scanner-specific signal characteristics. To validate this, we visualize the encoder's feature space using t-SNE. As shown in Figure 4(a), embeddings from different scanners form distinct clusters: Fuji Xerox ApeosPort C2060, Canon imagePRESS C650, and Canon imageRUNNER ADVANCE 6265 (which share similar error profiles) appear close together, while Plustek OpticBook 4800, exhibiting a distinct intensity–error response (Fig. 4(b)), forms a separate cluster. This confirms that the encoder captures scanner-specific variations critical for accurate correction.

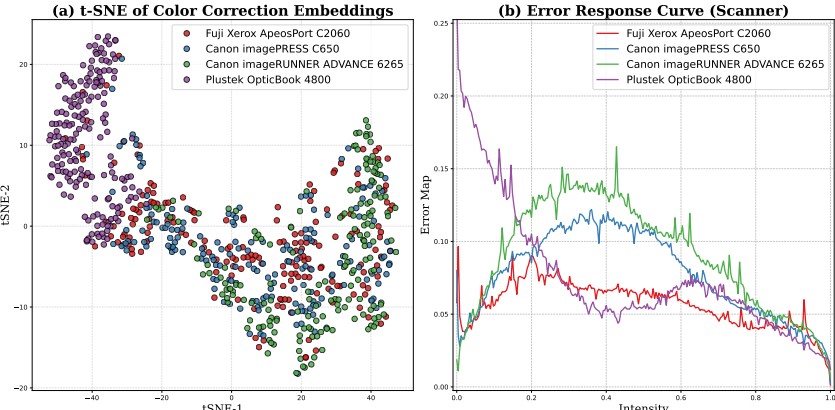

Figure 4: Encoder embeddings and intensity–error profiles. (a) t-SNE of global encoder features shows clear clustering by device, indicating that the encoder captures scanner-specific characteristics. (b) Intensity-dependent error response curves reveal non-linear, device-specific distortions across the intensity range.

## A5.2  PER-MODEL INTENSITY-DEPENDENT ERROR ANALYSIS

Figure 5 shows the per-model intensity-dependent error analysis. Among all methods, ImpRes-Descan achieves the lowest overall error while suppressing intensity-dependent distortions. Its error curves are smoother and more stable than those of DescanDiffusion+ and DocRes, which still exhibit residual intensity-dependent fluctuations. This demonstrates that our approach yields both minimal error and consistent correction across intensity levels.

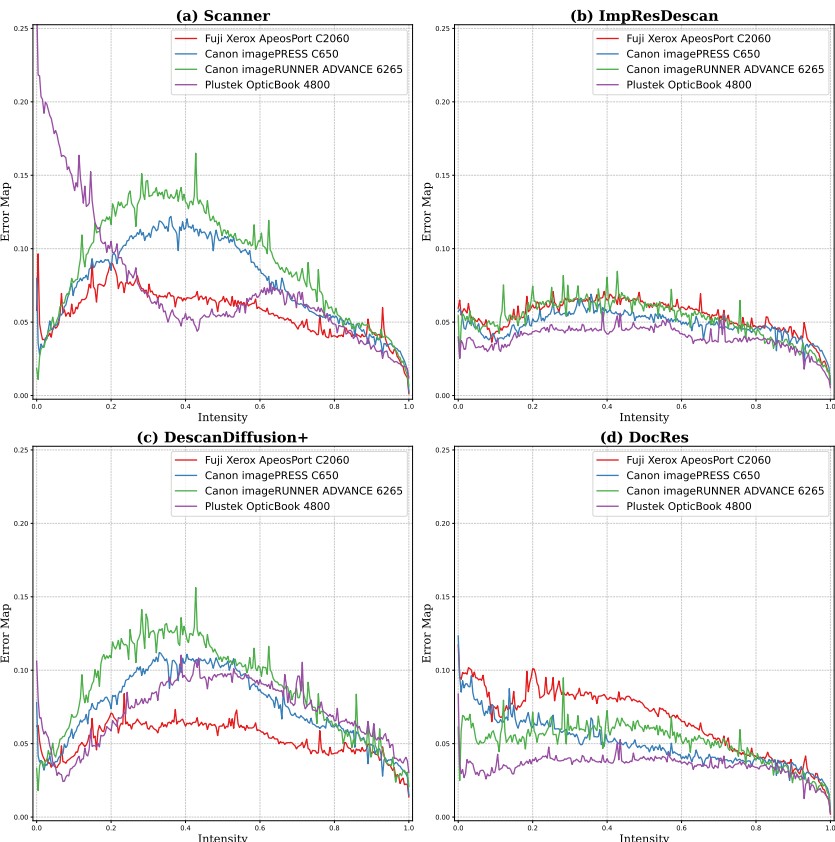

Figure 5: Per-model intensity-dependent error analysis. (a) Raw scanner error curves show strong, device-specific intensity bias. (b) ImpResDescan yields the lowest and most uniform error across intensities. (c) DescanDiffusion+ reduces error but retains residual intensity-dependent fluctuations. d) DocRes lowers the overall error but shows only moderate stability compared to the raw scanner outputs.

## A6 ADDITIONAL QUALITATIVE RESULTS

In this section, we present additional qualitative results for the different models. It can be observed that ImpResDescan is able to faithfully restore and recover fine color details. These examples include results from the DESCAN-18K dataset (Figure 6), as well as samples from the Comic (Figure 7) and DPS datasets (Figure 8).

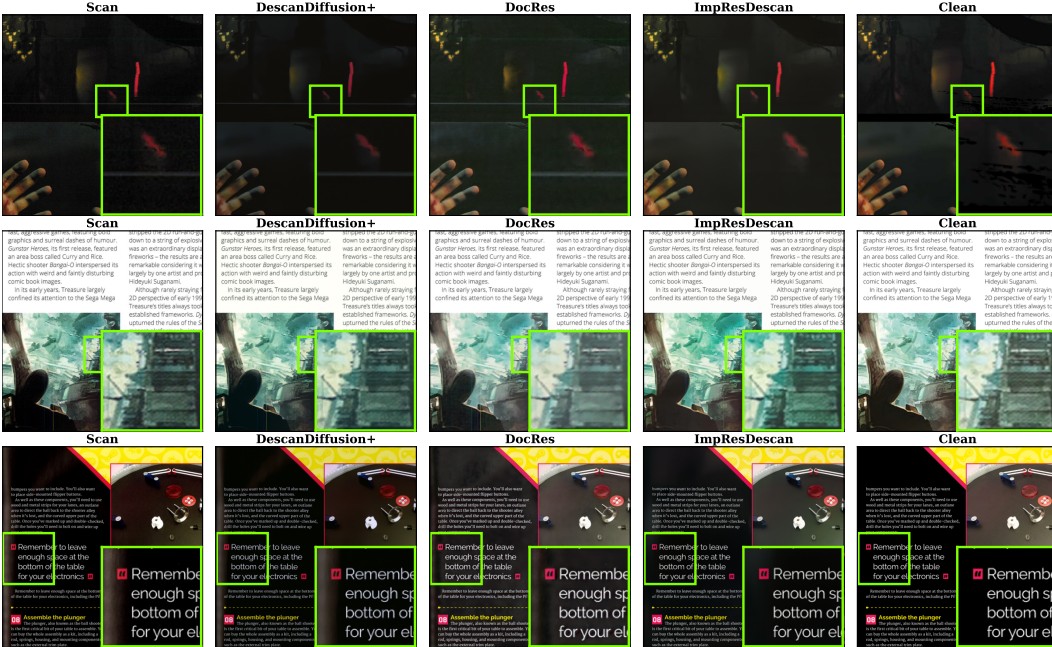

Figure 6: Each row shows, from left to right: **Scan**, **DescanDiffusion+**, **DocRes**, **ImpResDescan**, and the **Clean** digital original. Green boxes indicate regions used for zoomed inspection. Across diverse content, ImpResDescan shows preserved edges and reduced artifacts, including streaks, spine defects, and bleed-through.

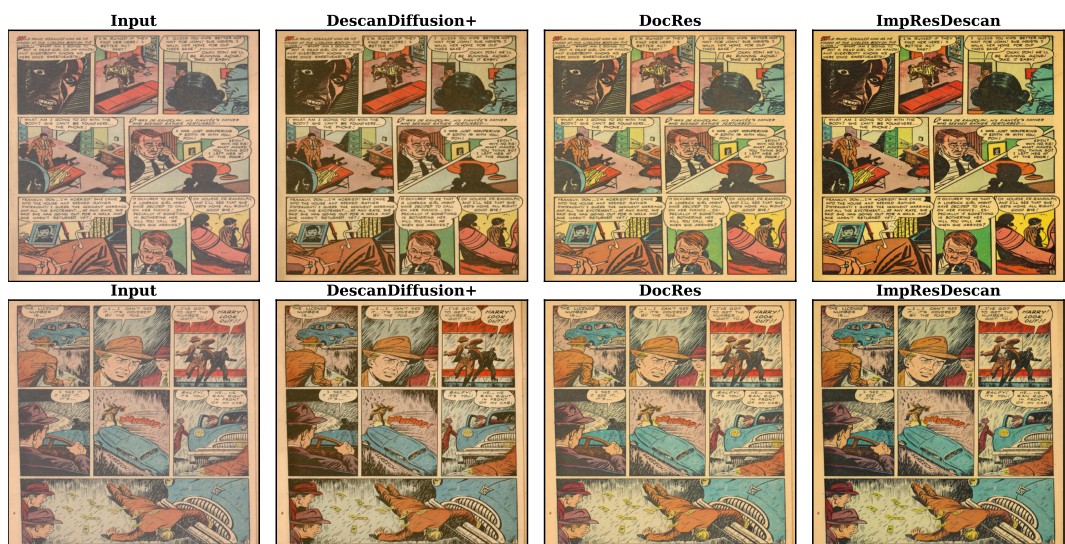

Figure 7: Non-reference qualitative comparison on Comic dataset. Each row shows, from left to right: **Input**, **DescanDiffusion+**, **DocRes**, and **ImpResDescan**. ImpResDescan produces sharper details, clearer text, and more faithful color reproduction.

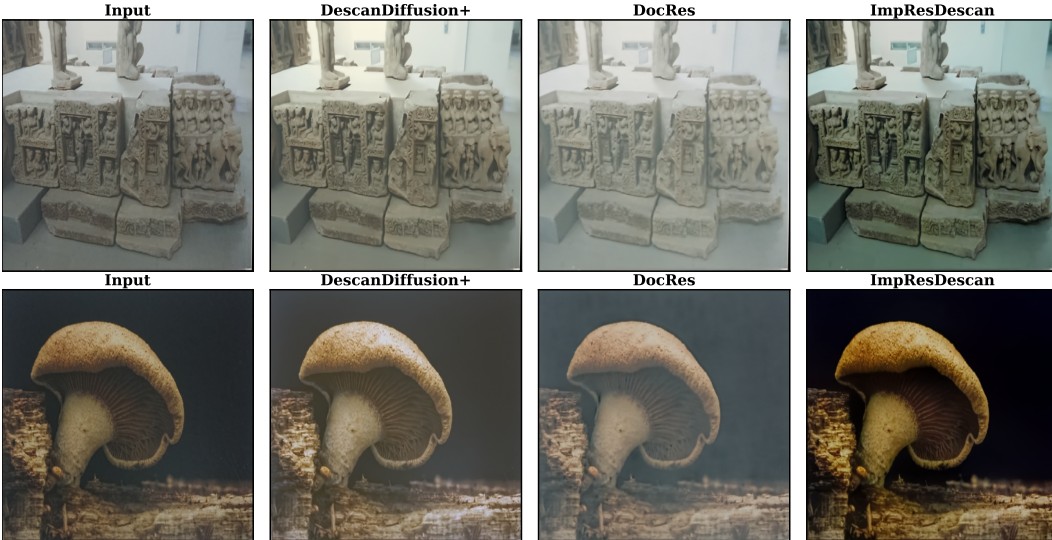

Figure 8: Non-reference qualitative comparison on the DPS dataset. Each row shows, from left to right: **Input**, **DescanDiffusion+**, **DocRes**, and **ImpResDescan**. ImpResDescan produces sharper details and more faithful color reproduction.

