# OpenReview forum: "ImpResDescan: Diffusion-Based Restoration for Scanned Document Images via Implicit and Ambient Training"
_ICLR.cc/2026/Conference — Submitted to ICLR 2026_

### Official Review · Reviewer_p5WF · 2025-10-28

**Soundness:** 3
**Presentation:** 3
**Contribution:** 2
**Rating:** 4
**Confidence:** 4

**Summary:**

This paper proposes ImpResDescan, a diffusion-based descanning framework that aims to restore digital-quality images from degraded scanned inputs. The method introduces two components: (1) an implicit color correction module that learns nonlinear color mapping without handcrafted priors, and (2) a degradation-aware local refinement module trained with ambient supervision and a multiscale sliced Wasserstein distance (MSSWD) loss to improve robustness under spatial misalignment. Experiments on the DESCAN-18K dataset show moderate improvements over DescanDiffusion+ in PSNR and FID.

**Strengths:**

1. The paper is clearly written and easy to follow.
2. The idea of using keypoint-based spatial masking for misalignment robustness is reasonable.
3. The authors report consistent improvements on multiple metrics.

**Weaknesses:**

1. Incremental novelty. The overall framework is largely based on DescanDiffusion+. The proposed two modules, implicit color
correction and ambient refinement, can be viewed as relatively minor architectural modifications on top of that pipeline. The core
diffusion-based restoration framework and residual training scheme are already established in the literature.
2. Although the paper claims improved robustness to spatial misalignment, it provides no quantitative or visual analysis (e.g., shift
map or displacement field visualization) to show reduced alignment error between restored and reference images.
3. The experiments heavily rely on DESCAN-18K, which was introduced in previous work but, this dataset is still not publicly
released. Without dataset access, reproducibility and fair benchmarking are difficult to verify.
4. While the paper includes evaluations on the Comic and DPS datasets, these two datasets alone may not fully demonstrate the
generalization ability of the proposed method. The authors are encouraged to further evaluate their approach on additional
publicly available document- or image-restoration datasets to strengthen the empirical evidence.

**Questions:**

1. While the paper includes evaluations on the DESCAN-18K, Comic, and DPS datasets, the external experiments mainly use noreference
metrics (LIQE, PAQ2PIQ, MUSIQ, Q-ALIGN) and involve domains (e.g., comics, camera-captured photos) that differ
from real scanned documents. It would be helpful to see additional experiments on publicly available document-restoration datasets.
2. The implicit color correction module functions like a general pre-processing stage and could in principle be replaced by any
published color-correction method. Since the paper does not compare against existing pre-processing approaches, it remains unclear
whether the proposed implicit color correction module provides substantial advantages beyond existing approaches.
3. In addition, could the authors discuss whether this implicit color correction can be directly integrated into other existing document
-restoration frameworks (e.g., DocRes, Restormer, or DescanDiffusion+) to yield similar benefits?

---

### Official Review · Reviewer_EeF9 · 2025-10-30

**Soundness:** 2
**Presentation:** 2
**Contribution:** 2
**Rating:** 2
**Confidence:** 4

**Summary:**

The paper presents a two phases framework (color correction and then refinement) for enhancing scanned images, based on supervised learning using a dataset of (original,scan) pairs.

**Strengths:**

The proposed method demonstrates performance improvements over a previous de-scanning technique and several more general methods.

**Weaknesses:**

- The proposed method is based on integrating exiting techniques in order to solve the specific de-scanning task. I did not observe any novel core insight in the paper.
This work may be more suitable to an application-oriented conference than to a general ML conference.

- It is not clear to me how scanning can lead to textual content misalignment (rather than just visually degraded content) as in Figure 1 left. If there are wrong pairs in the training set (hopefully not in the test set), examine ways to detect and remove them before training.

- Regarding spatial misalignments, in Figure 1 right, the misalignments look fixed and small.
Notice that if the misalignment is spatially fixed then it can be easily estimated using FFT, and if it is spatially varying but rather small (as expected when scanning) then it can be addressed with simple existing methods (e.g., used for image registration and optical flow). I believe that such alignments of the training data will allow to tackle the problem with a training strategy that is simpler than the approach proposed in the paper (constructing masks and ambient training).

- In Table 1, the performance gain over the more general method DocRes is not large.

**Questions:**

- The two summed terms in equation 8 have equal terms (d_ij-t_ij)^2 = (t_ij-d_ij)^2, so the formula is strange.

- Did you use the DESCAN-18K training set (used by your method) to enhance/fine-tune the more general image restoration methods?

---

### Official Review · Reviewer_5s58 · 2025-11-01

**Soundness:** 2
**Presentation:** 2
**Contribution:** 2
**Rating:** 4
**Confidence:** 3

**Summary:**

This paper introduces ImpResDescan, a novel framework for restoring images degraded by the print-scan process. The authors identify key limitations in prior state-of-the-art methods like DescanDiffusion+, which rely on simplistic assumptions such as linear color correction and perfect geometric alignment. To address this, ImpResDescan proposes a two-part, distortion-aware solution. Extensive experiments on datasets like DESCAN-18K demonstrate that ImpResDescan achieves new state-of-the-art performance, consistently outperforming existing models in both fidelity (PSNR) and perceptual quality (FID, LPIPS).

**Strengths:**

1. The experimental comparisons with related methods are thorough and comprehensive.
2. The motivation for the work and the overall approach are clearly articulated and easy to understand.

**Weaknesses:**

1. The meaning of 'res' in Figure 2 is not explicitly defined. The authors should clarify what it represents—for instance, does it refer to the intermediate layers of the diffusion network?
2. As shown in Table 3, the final proposed method exhibits a slight drop in SSIM performance. The paper would be strengthened by providing more visualizations and analysis to explain or justify this trade-off.
3. The paper would greatly benefit from a visual ablation study. Providing visual comparisons that incrementally add each component of the proposed framework would clearly demonstrate the specific contribution and effect of each part.
4. Which specific diffusion base model was used? The paper should also discuss how the choice of the base model affects the results. Would a better or larger model, like a flow matching architecture, improve the performance and speed?

**Questions:**

It is suggested to add hyperlinks in Table 1 for the different methods to make it easier for readers to read and compare.

---

### Meta-Review · Area_Chair_pXoa · 2026-01-05

**Summary:**

This paper proposes ImpResDescan to enhance scan images using a two-phase framework. All reviewers find this work interesting and provide some interesting results, yet reviewers are generally concerned about the novelty and evaluation of the proposed method. While authors do not respond to those concerns, the area chair recommends rejection.

**Reviewer Concerns:**

No rebuttal is provided.

**Reviewer Scores:**

No rebuttal is provided.

---

### Decision · Program_Chairs · 2026-01-26

Reject